# Phytotherapy of Vulvovaginal Candidiasis: A Narrative Review

**DOI:** 10.3390/ijms25073796

**Published:** 2024-03-28

**Authors:** Natalia Picheta, Julia Piekarz, Oliwia Burdan, Małgorzata Satora, Rafał Tarkowski, Krzysztof Kułak

**Affiliations:** 1Student’s Scientific Association at the I Chair and Department of Gynaecological Oncology and Gynaecology, Medical University of Lublin, Staszica 16 Str., 20-081 Lublin, Poland; natalia.picheta2812@gmail.com (N.P.); piekarzjulia1@gmail.com (J.P.); oliwia.burdan_in@interia.pl (O.B.); msatoraa@gmail.com (M.S.); 2I Chair and Department of Gynaecological Oncology and Gynaecology, Medical University of Lublin, Staszica 16 Str., 20-081 Lublin, Poland; rafaltar@yahoo.com

**Keywords:** VVC, phytotherapy, *Fridericia chica*, *Allium jesdianum*, *Curcuma longa*, *Cannabis savita* subs, *Anethum graveolens*, flavonoids

## Abstract

Vulvovaginal candidiasis (VVC) is a real gynecological problem among women of reproductive age from 15 to 49. A recent analysis showed that 75% of women will have an occurrence at least once per year, while 5% are observed to have recurrent vaginal mycosis—these patients may become unwell four or more times a year. This pathology is caused in 85–90% of cases by fungi of the *Candida albicans* species. It represents an intractable medical problem for female patients due to pain and pruritus. Due to the observation of an increasing number of strains resistant to standard preparations and an increase in the recurrence of this pathology when using local or oral preferential therapy, such as fluconazole, an analysis was launched to develop alternative methods of treating VVC using herbs such as dill, turmeric, and berberine. An in-depth analysis of databases that include scientific articles from recent years made it possible to draw satisfactory conclusions supporting the validity of herbal therapy for the pathology in question. Although phytotherapy has not yet been approved by the Food and Drug Administration, it appears to be a promising therapeutic solution for strains that are resistant to existing treatments. There is research currently undergoing aimed at comparing classical pharmacotherapy and herbal therapy in the treatment of vaginal candidiasis for the purpose of increasing medical competence and knowledge for the care of the health and long-term comfort of gynecological patients.

## 1. Introduction

Vulvovaginal candidiasis (VVC) is the second most common infection of the genital tract in women, right after the bacterial etiology of this inflammation [1]. It is characterized by burning, redness, vaginal swelling, and possible serous or watery vaginal discharge. In addition, pain during intercourse and dysuria are observed [2]. There is a distinction made for recurrent vulvovaginal candidiasis (RVVC), defined as the occurrence of a minimum of three episodes of the disease within a year [3]. As a result of this disease, patients experience increased stress, avoidance of sexual activity, and decreased self-esteem and confidence; around 53% of women with vaginal candidiasis have been diagnosed with depression [4,5].

Approximately 75% of the female population have experienced vaginal candidiasis at least once in their lives, while more than 50% of patients experience a second episode [6,7]. About 10–15% of women are found to be asymptomatically colonized by *Candida* [8]. Moreover, it is noted that the incidence of VVC apparently increases after the age of 17 [9]. On the other hand, by the age of 25, as many as 54.7% of patients have had issues with VVC [9]. RVVC is encountered in 9% of women between the ages of 25 and 49 [10]. It is roughly estimated that RVCC will affect, over the course of a lifetime, as many as 492 million women, while 138 million patients are affected annually [11].

A number of factors have been found to influence the likelihood of vulvovaginal candidiasis, including frequent antibiotic use, overly fitted clothing, and untreated diabetes [12]. A predisposition is also seen in people with iron deficiency anemia, immunosuppression, obesity, and menstruation [13]. Moreover, the appearance of thrush is influenced by high estrogen levels occurring in the following situations: hormone replacement therapy, pregnancy, and contraception [14]. Multiple instances of casual sexual intercourse are also a risk factor [15].

*Candida* yeasts belong to the natural physiological flora of the vagina, but in a situation of pathological growth and numerous virulence factors, inflammation develops [16,17]. The most common fungus responsible for the occurrence of VVC is *Candida Albicans*, which is responsible for 85–90% of cases [6,18,19]. It is usually observed in pregnant and premenopausal women with no comorbidities. Non-albicans Candida (NAC) species, which include *C. glabrata*, *C. dubiniensis*, *C. Tropicalis*, and *C. Krusei*, are also responsible for the occurrence of VVC [20]. *Candida glabrata* is the second most common agent responsible for vaginal candidiasis [8]. It is more commonly seen in diabetic and postmenopausal patients [21].

The diagnosis of VVC is based on the observation of clinical symptoms and microbiological examination, such as a culture with an antibiogram, which is still considered the gold standard for detection. While other methods have relatively poor sensitivity, cultures are often associated with delayed diagnosis and treatment [22].

More and more strains are becoming resistant to existing treatments, as drugs such as fluconazole are available without a prescription. Through easy availability, patients are increasingly turning to medicines that do not require medical professional services. Increasing resistance to antifungal drugs and an increase in predisposition to VCC are forcing researchers to look for new alternative treatments [23]. One of these is phytotherapy, which is based on the topical use of numerous herbs. The WHO reports that more than 80% of the population uses plant extracts, or active substances produced by plants, for therapy [24]. These include oils from: Roman chamomile (*Anthemis nobile*), fennel (*Foeniculum vulgarae*), lemon balm (*Melissa officinalis*), sage (*Salvia officinalis*), garlic (*Allium* L.), ginger (*Zingiber officinale*), nigella sativa, and fenugreek (*Trigonella foenum-graecum*). The best effect was noticed after using thyme and pine oil, which caused growth inhibition in 90% of *C. albicans* cells. In addition, great results in all tested species of the fungi were noted with the use of fennel oil [25]. In this paper, the authors will focus on the mechanisms and effects of several herbs in the treatment of VVC.

## 2. Materials and Methods

The keywords used for the database exploration included “vulvovaginal candidiasis”, “phytotherapy”, “garlic”, “berberine”, “CBD”, ”turmeric”, ”buyer”, and ”fridericia chica”. A search in the timeframe 2013–2023 provided 7324 papers. The PubMed database found 746 works, while 4485 were found in Scopus, and 2150 papers were sourced from Google Scholar. Of all the works found, only full text papers were considered. We included 78 papers, with the following inclusion criteria: original papers, papers written in English, and clinical cases. The exclusion criteria were review articles, articles not written in English, abstracts from conferences, and duplicated papers.

## 3. Results

### 3.1. Pathogenesis

*Candida* yeast enters the vaginal lumen and secretions mainly from the adjacent perianal area, which is located anatomically nearby. The presence of protective mechanisms against yeast enables candidiasis to persist as a commensal organism for a longer period in the avirulent phase [18]. VVC represents a symbiotic pathological condition caused by the excessive growth of *Candida* microorganisms, whose morphogenetic transitions to mycelium constitute the opening stage in pathogenesis [20]. Several decades ago, it was thought that *Candida* yeasts passively participate in the emergence of an opportunistic infection in a state of immunodeficiency in the host organism. The current knowledge is that these yeasts actively participate in the pathogenesis of the pathological process using mechanisms of aggression called virulence factors. Thus, the pathogenicity of *Candida* is influenced by multiple virulence factors, which chronologically include adhesion, biofilm formation, extracellular production of hydrolytic enzymes, filament formation, and, finally, phenotypic change. Figure 1, shown below, shows the process of the pathogenesis of candidiasis chronologically. Accurate knowledge of the pathomechanism allows the development of therapies that stop the cascade at a given stage to prevent the development of the disease. This is also important in the use of herbs that can specifically act on a special trigger point in the pathogenesis of candidiasis [8].

### 3.2. Treatment

Pathologies such as VVC can be classified as uncomplicated or complicated. This division depends on the factors of the severity of the infection, the species of fungus, and the natural predisposition of the patient’s immune system. It has been proven that complicated VVC less often responds positively to the proposed schematic therapy and requires radical methods. This provides motivation to take different directions in planning therapy using, among other things, herbs [22]. The gold standard for VVC treatments is azole-like medicines. They act on the ergosterol pathway, which is the building block of the fungal cell wall. Fluconazole is a commonly used agent from this group of drugs. This agent binds to the active site of the enzyme lanosterol–14–α–demethylase, thus terminating further reactions in the ergosterol pathway. It is worth noting that azole drugs have limitations in the nature of causing numerous side effects such as headache, dizziness, and nausea and a noticeable tendency for *Candida* to acquire resistance to these drugs [26]. Figure 2 shows the currently accepted gold standard of VVC treatment. It presents that with the persistence of symptoms, an increasingly prolonged use of fluconazole is recommended, which, as discussed above, is not an optimal solution for patients and exposes them to the occurrence of troublesome side effects, for example, hepatotoxicity, fatigue, myalgia, fever, malaise, torsade de pointes, and QT prolongation, although they are rare [27,28].

Therapeutical difficulties, such as recurrence of disease, motivate the development of new medicines for VVC. Oteseconazole is a pharmaceutical product belonging to the group of new azole drugs. It is a metalloenzyme inhibitor targeting fungal sterol demethylase 14α [cytochrome P450 (CYP) 51 (CYP51)]. As a result of its inhibitory effect on CYP51, the enzyme, which is involved in ergosterol synthesis, causes the accumulation of 14–methylated sterols, some of which are toxic to fungi [29]. Unlike other azoles, which contain an imidazole or triazole grouping that binds the human cytochrome, oteseconazole has a tetrazole grouping. This makes its target specificity toward the ergosterol synthesis substrate higher. Studies to date of oteseconazole in vivo and in vitro have shown the potential for fewer side effects than previous-generation azole antifungals, but it is not the perfect opportunity [30]. In VVC therapy, after thirty years of the clear dominance of azole drugs, a completely innovative first-in-class oral triterpenoid-glucan synthase inhibitor ibrexafungerp has emerged [31]. The drug ibrexafungerp exhibits in vitro fungicidal activity against various strains of *Candida* species. It has been shown to be effective against strains resistant to widely used echinocandins and azoles. Studies with ibrexafungerp in 449 patients have shown enhanced antifungal activity against *C. albicans* at a vaginal pH (4.5) consistent with candidiasis infections, compared to the standard laboratory pH level of 7.0. This represents a distinguishing feature of the medicine as a pharmaceutical [32]. Additional studies have also shown a high potential for ibrexafungerp to accumulate in tissues and vaginal secretions, which represents a chance for a longer therapeutic effect of the drug [32]. Unfortunately, despite its huge number of advantages, it is not free from side effects. Negative consequences recorded in women using ibrexafugerp include gastrointestinal disorders in 14.8% of patients [32]. The treatment of VVC continues to be a major therapeutic target, entailing many unmet challenges in terms of drug therapy, for which herbal medicine may be an alternative in the future.

### 3.3. Effects of Plant Metabolites on Fungal Infections

Plants contain numerous substances, called secondary metabolites, with potential therapeutic effects. Phenolic compounds, or phytochemicals, in plants are mainly responsible for the color of leaves and fruits. They show interactions with the membrane and cell wall of fungi and nucleic acids. In addition, they inhibit the transport of energy in the cell and, thus, its functions, and affect the binding of carbohydrates, making it difficult for fungal cells to access them [33].

Flavonoids are compounds found in vegetables, fruits, and the stems and flowers of plants. They have anti-inflammatory, anti-allergic, diuretic, and spasmolytic effects. The flavonoid mechanism used in antimicrobial activity is related to the inhibition of nucleic acid synthesis; the B ring, contained in the structure of flavonoids, affects the formation of hydrogen bonds between nitrogenous bases [34]. The second mechanism is the inhibition of DNA gyrase, in whose ring hydroxylation is present, while luteolin and quercetin, which are flavones, inhibit topoisomerase I and II [25,35]. Other compounds, namely alkaloids, one of which is berberine, inhibit biofilm formation [36]. Glucosinolates are compounds that gain antimicrobial properties only after conversion to isothiocyanates or nitriles. They cause a change in the polarity of the cell membrane, resulting in its disruption [25].

Other important compounds are terpenoids, and in vitro studies have shown that carvacrol, eugenol, and thymol cause hydrogen and calcium ions to flow out of cells, thereby disrupting the TOR signaling pathway; in a study conducted in vitro on thirty-six clinical isolates of *C. albicans* and four other strains (*C. parapsilosis*, *C. blankii*, *C. kefer*, and *C. pseudotropicalis*), eugenol caused the death of 99% of fungal cells on the media [37]. Other likely mechanisms are blocking the cell cycle at specific stages, inhibiting *C. albicans* adhesion, or degrading fungal cell walls [38].

### 3.4. Herbal Preparations Used to Treat the Vagina and Vulva

*Candida albicans* has several virulence genes that predispose it to causing vaginal and vulvar infections, one of which is the *SIR2* gene (*silent mating–type information regulation 2 gene*). Its deacetylation leads to the production of sirtuins, which are essential for the transition from yeast to a filamentous form [39,40]. A decrease in *SIR2* gene expression was observed during exposure to fresh garlic extract containing allicin; the relative quantification of *SIR2* gene expression in the control group untreated with the tested substance was approximately 0.8 (ratio of the gene of interest/actin). After treatment with allicin at doses of 20, 40, 60, 80, and 100 mg/mL, it decreased to approximately 0.5, 0.6, 0.65, 0.35, and 0.3, respectively [41]. Another virulence factor of *C. albicans* is the *ECE1* gene (*endothelin converting enzyme 1 gene*), which encodes a peptide toxin, candidalysin, which, by destabilizing the cell membrane of vaginal cells, facilitates infection and is responsible for the immunopathogenesis of fungal VVC [42]. An increase in its synthesis is particularly evident with the proliferation of filaments [43,44]. It was tested whether *ECE1* responded equally well to the ingredients contained in garlic. Although this was only a preliminary study, during allicin treatment of *C. albicans* clinical isolate 0861 at 10 μg/mL and 60 μg/mL, the relative quantification of *ECE1* expression dropped almost to 0 (the relative quantification of *ECE1* gene expression in the control group was 1) [45]. The effect of ornamental garlic on fungal infections with the second most common etiological agent, *C. glabrata*, was also investigated. A study was conducted in which a clinical isolate of *C. glabrata* was collected from 32 patients with VVC and RVVC. An aqueous ethanolic solution of *Allium jesdianum* inhibited the growth of fungal colonies both with infections with azole-susceptible and azole-resistant strains, but the mechanism is not further understood [46]. Table 1 shows the minimal inhibitory concentration (MIC) values for fluconazole and *Allium jesdianum* extract. 

Based on the data presented in Table 1, there is a clear difference between the MIC values; the lower MICs for *Allium jesdianum* in both tested values (50% and 90%) means that a lower concentration of the substance is needed to effectively control the infection compared to fluconazole.

Another plant with a potential therapeutic effect in vaginal candidiasis is *Fridericia chica*. This plant is rich in many compounds that exhibit anti-inflammatory and antifungal properties, i.e., flavonoids, alkaloids, terpenoids, and steroids, but the first category are mainly responsible for the desired therapeutic aspects [47]. So far, 39 types of flavonoids have been isolated, including 4′-hydroxy-3,7-dimethoxyflavone, kaempferol, and vicenin II, which possess anti-inflammatory and antifungal properties [48]. Therefore, it was decided to investigate whether *Fridericia chica* would be effective in the treatment of vaginal and vulvar mycosis of strains resistant to standard treatment. It was shown that a hydroethanolic extract of *F. chica* leaves (HEFc) delays fungal proliferation, so it could help the body to fight the infection on its own [48]. Another mechanism is the reduction of the yeast to filamentous form transition, which impairs the ability of pathogenic fungi to infect vaginal tissues. In the study, compared to a control group using nystatin at 8 μg/mL, *F. chica* leaf extract at 128, 256, and 512 μg/mL reduced the development of fungal filaments. At a dose of 1024 μg/mL, it had a fungicidal effect even though an increase in the transition of the fungi to an invasive form was found. However, this effect should not be confused with a decrease in viability, as the plant does not cause a decrease in *C. albicans* survival [49]. In an animal model that divided thirty-six rats randomly into six groups, vaginal administration of a hydroethanolic solution of *F. chica* leaves resulted in resolution of symptoms within 6 days, confirming its efficacy [49].

The families *Annonaceae: Rollinia* and *Xylopia; Berberidaceae: Berberis, Caulophyllum*, and *Mahonia; Menispermaceae: Tinospora; Papaveraceae: Argemone, Bocconia*, and *Chelidonium; Ranunculaceae: Coptis, Hydrastis* and *Xanthorhiza*; and *Rutaceae: Evodia, Phellodendron* and *Zanthoxyllum* are rich in berberine, but *Berberis vulgaris* is known for the highest concentration of this substance; it contains 8% of alkaloids, of which 5% are berberine [50]. It is a compound that has been known in Chinese medicine for a long time, and has been attributed with various properties, including anti-inflammatory and antioxidant properties [51]. It can also be used in gastrointestinal diseases, in anticancer therapy due to its antiproliferative effects, or in metabolic disorders by lowering cholesterol [52]. Due to its numerous applications, questions have been raised about its efficacy in the treatment of VVC.

The first step in infection, necessary for the occupation of vaginal cells and the start of the colonization process of *C. albicans*, is adhesion. One of the proteins associated with adhesion is ICAM-1 (Intercellular Adhesion Molecule 1). This is a protein whose expression increases in response to inflammatory reactions and in a high glucose environment, which is important in fungal infection [53]. By increasing the expression of ICAM-1, the fungus cells more easily adhere to the vaginal epithelium [54]. Other proteins involved in adhesion are mucins 1 and 4, the expression of which also increases during fungal infection and decreases when the infection subsides. For this reason, a good therapeutic solution could be to stop this process to inhibit further infection. Therefore, the effect of berberine on the adhesion of *C. albicans* strain SC5314 was investigated; a decrease in ICAM-1 expression was observed after application. After 1 h, the relative expression was 1.25 in the control group and 1.2 in the group with berberine, while after 3 h, it was 1.5 and 0.75, respectively; thus, the first stage of infection was significantly reduced [55]. Similar conclusions were made for mucin 1 and 4; the application of the berberine preparation resulted in down-regulation of the expression of these genes. For mucin 1, after one hour in the control group, the expression was 1.0, and in the group with berberine, it was 0.7; after 3 h, the expressions were 1.4 and 0.6, respectively. In contrast, for mucin 4, after 1 h, 1.1 and 0.7, respectively, and after 3 h, 1.6 and 0.8 [55].

A study involving a four-point assessment of the temporal efficacy of berberine showed that its best inhibitory effect on *Candida* biofilm growth and viability was seen after 48 h of application. When comparing the MIC values of berberine and fluconazole, it was found that they differed in effectiveness depending on the type of fungal strains. Some strains were extremely sensitive to berberine; for example, for *C. krusei* the MIC = 10, with a fluconazole MIC of 64. In contrast, *C. albicans* strains proved to be much more sensitive to antifungal drugs at MIC = 0.5, whereas the berberine MIC was much higher (MIC = 80/160) [36].

An herb worth looking into is *Curcuma longa*, otherwise known as long oyster, commonly known to all as a spice. In 2018, it was approved by the FDA (Food and Drugs Administration) as having a good safety profile [56]. It has antimicrobial, as well as antioxidant and anti-inflammatory effects, through its effects on Th-17 cells and regulatory lymphocytes. The former enhance the inflammatory process by increasing the synthesis of interleukins 17, 22, 23, while the latter calm this trial. It is therefore important to maintain a balance between the two in order to reduce the inflammatory effect and counteract related diseases. Turmeric inhibits Th-17 and ensures that the balance between Th-17 and Th-regulatory cells is maintained [57]. It is also credited with antifungal activity, so its effectiveness in vaginal infections has been tested. 

Ninety-four women with VVC were studied and divided into two groups: those taking a 10% vaginal cream with curcumin and a 1% clotrimazole-based vaginal cream. It was observed that vulvar and vaginal pruritus was reduced in both groups: after clotrimazole, from 87.2% to 19.1%; after curcumin, from 85.1% to 12.8% [58].

In addition, curcumin affects the survival of fungal cells by leaking intracellular potassium and damaging the cell membrane, thereby disrupting its integrity [59]. Another likely mechanism of action of curcumin is the inhibition of hydrogen ion release and the reduction of ergosterol production. This results in a decrease in proteinase secretion from fungal cells and destruction of their membranes [58].

An herb that causes a lot of controversy is *Cannabis sativa subs. Indica*. As is well known, cannabidiol (CBD) has a wide range of medical applications: it can be used to treat persistent vomiting in patients after chemotherapy, as an analgesic for people with chronic pain, and in multiple sclerosis, it has been proven to reduce muscle spasticity [60,61,62]. The efficacy of cannabidiol has been studied on *C. albicans* SC5314, which carries a green fluorescence protein known as CBD in VVC. It has been shown to contribute to the inhibition of fungal biofilm formation and removal at concentrations below minimum fungicidal and inhibitory concentrations [63].

This is a rather complex, multifaceted mechanism of action: first, it affects biofilm formation. We treated fungal cells with different concentrations of CBD and observed inhibition of *C. albicans* colony formation. At a dose of 6.25 μg/mL, it reduced biofilm development by 28% after 48 h, and by 39% after 72 h, compared to the control group not treated with CBD. At a dose of 25 μg/mL, the number of active *C. albicans* cells in colony formation decreased on subsequent days; the results are shown in Table 2 below [63].

Importantly, CBD destroys the biofilm by reducing the expression of genes encoding the production of fungal cell wall components, including exopolysaccharides (EPS), such as *ADH5* (*alcohol dehydrogenase 5*), the gene encoding alcohol dehydrogenase associated with cell wall building, by more than 3-fold; *ECE1* and *EED1* (*epithelial escape and dissemination 1*), needed for the proliferation of the filaments, by more than 6-fold; and *ALS3* (*agglutinin-like sequence 3*), which encodes a cell wall glycoprotein responsible for adhesion, by almost 2-fold (1.72) [63,64,65,66]. The expression of genes responsible for maintaining the integrity of the cell membrane was also downregulated 3-fold, which increases their permeability and helps fight infection [63]. The second mechanism of action involves the hyperpolarization of the mitochondrial membrane of fungus cells; consequently, the concentrations of ATP after CBD application at concentrations of 6.25, 12.5, and 25 μg/mL decreased by 80%, 85%, and 90%, respectively [63]. The third mechanism is an increase in the production of reactive oxygen species (ROS), which will cause increased cell death in *C. albicans* cells [60,67]. Based on the research results presented so far, cannabidiol looks the best compared to the other herbs presented. Using it in practice could speed up the process of fighting vaginal infections.

The first-line drug used in women is mainly fluconazole due to its broad spectrum of action. However, there is an increase in resistance to this drug due to, among other things, mutations in the *ERG11* gene (*sterol 14–demethylase*). The enzyme encoded by this gene, 14-alpha demethylase of lanosterol, will have a lower affinity for the drug; in turn, the *CDR* (an ABC transporter) and *MDR* genes (a major facilitator) are overexpressed, resulting in reduced accumulation of the drug in the fungal cell [68]. A study based in Africa have shown that herbalists use different methods and agree on high efficacy [69]. They have identified some 32 plant species with antifungal activity, of which 43.8% are herbs, including *Momordica foetida*, *Sansevieria dawei*, and *Clerodendrum umbellatum* [69]. Many of the herbs used in Uganda have not been described by the specialized literature, but *Momordica foetida* has been shown to contain alkaloids, flavonoids, and phenolic glycosides with potential medicinal properties [69]. It is a plant widely used in tropical Africa, and has antidiabetic, antilipogenic, and laxative properties, but the mechanisms of action are not known so far. It is a good antioxidant [70]. *Sansevieria dawei* has saponins, flavonoids, and terpenoids with antifungal properties, while *Sansevieria Hyacinthoides* has antifungal activity against *Candida albicans* [69]. *Khaya anthotheca*, a tree whose root extracts are used by herbalists, has shown potential activity against *C. krusei* [71]. Although none of these plants have been fully studied, the fact that they are very popular and effective in Uganda allows one to believe them as a promising alternative to classical pharmacotherapy.

The next and last plant discussed in this review is dill, *Anethum graveolens*, which is popular and used in Ayurvedic medicine. The Table 3 shows the compounds that each part of the dill contains [72].

Dill seed essential oil attacks the cell membranes of *C. albicans* and their mitochondria, thereby destroying the cells. The amount of ergosterol in cell membranes after treatment with dill seed oil was investigated; the average decrease in sterol content for different strains oscillated between 33% and 75% [73]. The following Table 4 shows the effect of different concentrations of dill seed essential oil on the amount of ergosterol in *C. albicans* strains.

The reduction of ergosterol leads to membrane destabilization and cell death. *Anethum graveolens* seed oil also inhibits mitochondrial dehydrogenase, an enzyme involved in ATP synthesis. The higher the concentration, the lower the activity. Table 5 below shows the concentration dependence of activity [73].

The study involved 60 women, divided into two groups differing in their VVC treatment regimen. Half of the patients, randomly selected, were treated with clotrimazole (control group), while the rest were medicated with *Anethum Graveolens* (study group). In both groups, no statistically significant difference was noted in the incidence of adverse reactions, culture results, and satisfaction with the therapy. Pruritus in the study group occurred in 20% of patients, while in the control group it occurred in 16.7%. Positive cultures were tested in 10% of women tested with dill-containing suppositories, while this result was 13.3% in patients treated with clotrimazole. Therapeutic satisfaction was demonstrated by 100% of control patients, while in the study group, it occurred in 93.4% [74]. 

Table 6 summarize the findings reported in the manuscript.

## 4. Discussion

The question that a modern medical specialty such as gynecology should ask itself in a manifestation of concern for the permanent well-being and high comfort of patients suffering from a disease such as vaginal and vulvar candidiasis is whether phytotherapy can represent the future in the treatment of VCC. Will herbal treatment one day be the standard therapy for VVC? Is phytotherapy capable of replacing classical pharmacotherapy?

VVC is a significant clinical problem affecting an increasing number of female patients. Due to the widespread availability of the most popular drugs from the azole group without a prior prescription from a specialist, they are increasingly used on their own and abused by patients, which contributes to the growing number of strains resistant to the existing treatments. Adverse symptoms, mainly from the gastrointestinal tract, such as vomiting, nausea, and diarrhea, are another problem, but convulsions, dizziness, and insomnia also occur [28]. For this reason, it is a good idea to look for natural methods; the purpose of this paper was precisely to present an alternative form of treatment with herbs, which is little talked about.

Fungal virulence is related to various factors and processes, such as adhesion, expression of genes, e.g., *ECE-1* or *ICAM-1*, or the transition of the yeast to filamentous form. The studies cited in this review clearly state the satisfactory efficacy of herbal preparations at all these stages, and present other actions such as inhibition of mitochondrial dehydrogenase activity. Comparisons of their action to classical pharmacotherapy also speak in favor of the alternative method; the effectiveness of herbal preparations is better than commonly used drugs. The most expressive example is curcumin. A complete cure of VVC in women using turmeric cream was observed in 66% of patients, whereas after clotrimazole, the result reached 48.9% [58]. This is probably caused by the fact that curcumin reduces the induction of multiple inflammatory mechanisms of cyclooxygenase 2 and interleukins Il-1, -5, -6, -12, and -18 and lox-12. On the other hand, berberine showed broad-spectrum antifungal activity, making it a new potential therapeutic agent and a primary active therapeutic substance [36]. 

In addition, the possibility of herbal resistance is very low, as they are natural methods and are uncommonly recommended in gynecological offices [75]. Nevertheless, using herbs and medicines at the same time increases the therapeutic effect [76]. Also, garlic has satisfactory results; during allicin treatment of C. albicans clinical isolate 0861 at 10 μg/mL and 60 μg/mL, the relative quantification of *ECE-1* expression dropped almost to 0 [73].

The herbs mentioned in this review offer new therapeutic options. If phytotherapy used in monotherapy would not yield the desired results, one could consider combining them with classical drugs as adjunctive preparations. To date, no studies confirm this, but perhaps the combination of classical antifungal drugs with, for example, curcumin affecting the fungal cell in the same way would achieve a synergistic effect. However, if it turned out that by the same mechanism, the effects achieved could be worse than in monotherapy, it would be possible to combine preparations with different effects in the cell, e.g., azole acting on ergosterol synthesis with CBD reducing ATP synthesis in the cell, which would intensify the therapeutic effect.

Almost all studies to date have been conducted in laboratory settings, and there is a lack of clinical studies on patients to confirm this efficacy, so it is worth expanding the research work. Some of the studies presented in the review lack specified sizes of the groups on which the herbal preparations were used, and, if specified, their numbers are small. In addition, the FDA still has not approved most of the herbs mentioned in the review, i.e., *F. chica* or CBD, but perhaps this will change with the increase in the number of studies on patients. On the other hand, the EMA (European Medicines Agency) approved, for example, *Curcuma longa* for relief of digestive disturbances such as slow digestion and flatulence [77]. There is still a lack of data regarding the use of phytotherapy in pregnant women, but also there is no information about its harmfulness [77]. What effect do the herbs mentioned in the review have on the fetus?

In addition, the use of honey and yoghurt also has good results. Allam M. et al. studied its effectiveness on pregnant patients (82 in the study group and 47 in the control group with tioconazole), where the clinical cure rate was higher in the study group (87.8%) than in the control group (72.3%). In addition, the incidence of side effects in the groups was adequate at 24.3% and 29.7%, respectively [78].

Phytotherapy is a thriving sphere of medicine. Herbs are an alternative to commonly used drugs for a growing number of conditions. However, research is still needed to confirm the actions of the herbs and other substances described in this work. Their use is an opportunity to develop new therapeutic methods. It would also be appropriate to focus on gynecological patients suffering from other diseases and aspects of the treatment of pregnant women and the effects of phytotherapy on the fetus.

## 5. Conclusions

Although phytotherapy and its use in daily gynecological practice require further clinical research, it appears to be a very promising therapeutic method. A review of the available literature has shown that herbs are as effective as classical treatment solutions. The use of herbal preparations will also not increase resistance to previously used drugs, and due to fewer side effects, patients may be more eager to accept their use. Numerous herbs mentioned in the review, which include garlic, *F. chica*, and berberine, inhibit the transition of fungi into an invasive form, and dill reduces the concentration of ATP in the fungal cell by inhibiting mitochondrial dehydrogenase. Turmeric, on the other hand, blocks ergosterol synthesis, but the best in this review is CBD, which has multiple mechanisms to limit fungal growth. This review provides evidence of the efficacy of phytotherapy and shows that it could become the future of VVC therapy.

## Figures and Tables

**Figure 1 ijms-25-03796-f001:**
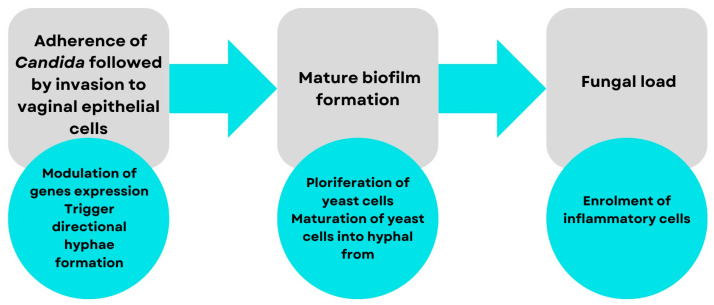
Pathomechanism of VVC [8]. Cell adhesion to the vaginal wall is followed by virulence, gene expression, and biofilm formation [8].

**Figure 2 ijms-25-03796-f002:**
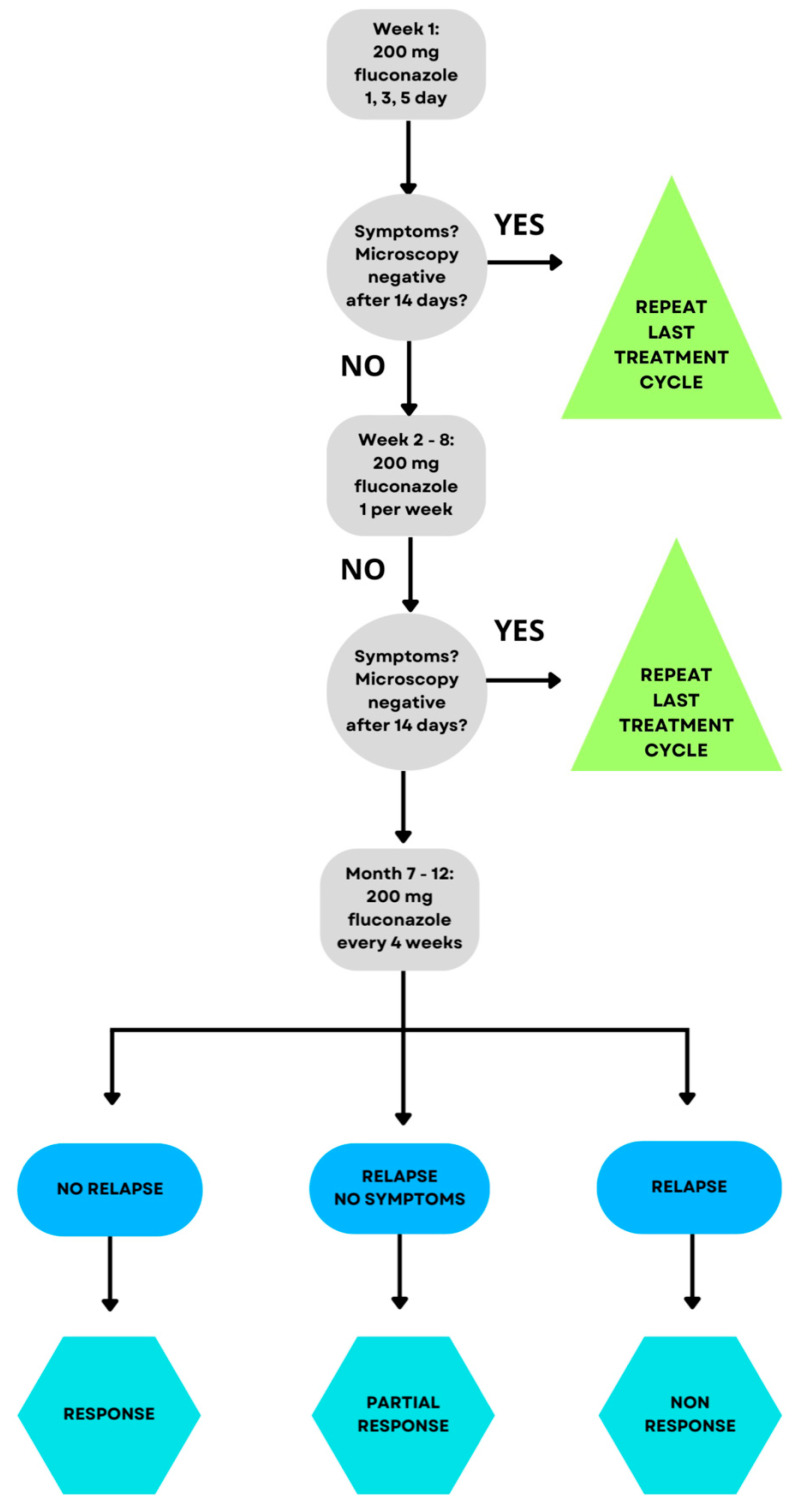
Treatment regimen for VVC [27,28]. Treatment begins with 200 mg of fluconazole; if symptoms persist and cultures are positive, 200 mg of fluconazole is given once a week for 6 weeks. If there is no improvement, 200 mg of fluconazole is administered once every 4 weeks for 6 months [27,28].

**Table 1 ijms-25-03796-t001:** Comparison of MICs for fluconazole and *Allium jesdianum* [46].

Drugs	MIC 50%	MIC 90%
Fluconazole	8 µg/mL	32 µg/mL
*Allium jesdianum*	2 mg/mL	3 mg/mL

**Table 2 ijms-25-03796-t002:** Number of active fungal cells in biofilms after application of CBD at a dose of 25 μg/mL [63].

Hours	Number of Active *C. albicans* Decreased Cells in Biofilms [%]
24	48
48	64
72	87

**Table 3 ijms-25-03796-t003:** Metabolites in different parts of garden dill [72].

Metabolites	Roots	Leaves	Seeds
Flavonoids	Present	Present	Present
Terpenoids	Present	Present	Present
Tannins	Present	Present	Present
Saponins	Present	-	Present
Cardiac-glycosides	Present	Present	Present

**Table 4 ijms-25-03796-t004:** Effect of fennel seed oil on ergosterol synthesis [73].

Strain of Fungus	Percentage of Ergosterol Synthesis Inhibition in *C. albicans* Cells at a Certain Concentration of Fennel Essential Oil [%]		
	0.078 μg/mL	0.156 μg/mL	0.312 μg/mL
*C. albicans* 09–5304	35.71%	42.85%	71.43%
*C. albicans* ATCC 64550	37.50%	45.83%	75.00%
*C. albicans* 09–1502	33.33%	41.67%	70.83%

**Table 5 ijms-25-03796-t005:** Effect of fennel seed oil on mitochondrial dehydrogenase activity [73].

Concentration [μg/mL]	Activity [%]
0.078	98.89
0.156	91.46
0.312	86.69
0.625	75.04
1.25	55.01
2.5	45.87
5	38.28
10	28.37

**Table 6 ijms-25-03796-t006:** Discoveries of authors included in this overview.

Author	Year	Substance	Searching Group	Main Findings
C. F. Low [41]	2008	Allicin	*C. albicans* ATCC isolated was treated with different concentrations of garlic extract	Reduction in *SIR2* gene expression a decrease in invasive form of the fungus
Y. Chen [73]	2013	Anethum graveolens seed oil	Isolates from patients vagina *C. albicans* 09–1502 and 09–5304 were incubated with essential oil from dried dill seads	Inhibition of mitochondrial dehydrogenase a decrease in ATP levels in the fungal cell
N. Abouali [58]	2019	Curcumin	Randomized study trial on 94 women– they were randomly assigned into two groups–fist one use curcumin–based 10% vaginal cream and second 1% vaginal clotrimazole cream	Reduction in ergosterol synthesis
M. M. Said [45]	2020	Allicin	*C. albicans* ATCC 14053 and two clinical isolates of *C. albicans* 0861 and 1358 was treated of freshly prepared garlic extract	Down-regulation of expression of *ECE1* gene encoding candidalysin a inhibits the proliferation of the filaments
M. Feldman [63]	2021	CBD	*C. albicans* SC5314 and *C. albicans* SC5314 with green fluorescence protein–control group were treated with ethanol and blank samples were with CBD	Decrease in expression of *ADH5* gene encoding alcohol dehydrogenase a decrease in cell wall synthesis. Increase in ROS levels a increase in fungal cell death. Mitochondrial membrane hyperpolarization a decrease in ATP concentration.
W. G. Lima [49]	2022	Hydroethanolic extract from *F. chica* leaves	After infection rats were randomly assigned to 6 groups of 6 each: fist–non– infected, second– infected and untreated, third and the rest infected and treated with hydroethanolic extract from *F. chica* leaves with different concentrations	Inhibiting the transition of the fungus into an invasive form
T. Zhao [55]	2022	Berberine	*C. albicans* SC5314 was treated with berberine	Limiting the transition of fungi to an invasive form

## Data Availability

Not applicable.

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
