# Peer review of "Phytotherapy of Vulvovaginal Candidiasis: A Narrative Review"

_ijms, 2024, doi:10.3390/ijms25073796_

Round 1

Reviewer 1 Report

Comments and Suggestions for Authors

The review discussed and explored using of alternative treatment, physiotherapy, in a common gynecological disease that used to be treated with chemical anti fungal. The authors covered all points about the pathogenesis and the mechanism of each plant extract. I have some minor comments:

1- page 3 line 106.. Figure number 2 not 1

2- page 4 line 127...figure number 3 not 2 

3- in the flowchart figures, please replace the dark blue background with lighter color because it is immpossible to read what inside the boxes in hard printed copy.  

Author Response

Manuscript: ijms - 2926187

Response to the reviewers

Dear Reviewer,

First and foremost, thank you for your time that You dedicated to prepare the comments to our manuscript entitled: "Phytotherapy of vulvovaginal candidiasis - a narrative review”. We consider all of the comments very valuable and helpful for the improvement of our paper. We have studied each comment very carefully and made appropriate corrections, which we hope will meet Your approval. Main corrections in the paper and responses to the reviewer’s comments are listed below.

On Behalf of the authors,

Małgorzata Satora

#Review 1 - the corrected passages, as suggested, are highlighted in pink in the manuscript.

  1. Page 3 line 106.. Figure number 2 not 1

Answear: Thank you for your valuable comment! However, after an in-depth analysis, we came to the conclusion that we would remove 1 figure, which carried with it the lack of change in the suggested figures, which would have made the most sense if figure 1 had remained in the article. Of course, all the data that were in figure 1 were included in the text.

  1. Page 4 line 127...figure number 3 not 2

Answear: Thank you very much! According to the response to your first comment, the changes we have made have caused figure 2 to remain figure 2 after all.

  1. In the flowchart figures, please replace the dark blue background with lighter color because it is impossible to read what inside the boxes in hard printed copy.  

Answear: Thank you for the right comment! In accordance with it, we have changed the background color in Figure 1 and 2 to a lighter color and the font to a more readable one.

Reviewer 2 Report

Comments and Suggestions for Authors

The use of plant medicine therapy for the treatment of vulvovaginal candidiasis is a relatively ambitious proposition. There is not much connection between plant medicines, and the overall review does not appear to be very organized, logical, or readable. There are individual spelling or omissions in some parts of the article.

Author Response

Manuscript: ijms - 2926187

Response to the reviewers

Dear Reviewer,

First and foremost, thank you for your time that You dedicated to prepare the comments to our manuscript entitled: "Phytotherapy of vulvovaginal candidiasis - a narrative review”. We consider all of the comments very valuable and helpful for the improvement of our paper. We have studied each comment very carefully and made appropriate corrections, which we hope will meet Your approval. Main corrections in the paper and responses to the reviewer’s comments are listed below.

On Behalf of the authors,

Małgorzata Satora

Review 2

  1. The use of plant medicine therapy for the treatment of vulvovaginal candidiasis is a relatively ambitious proposition. There is not much connection between plant medicines, and the overall review does not appear to be very organized, logical, or readable. There are individual spelling or omissions in some parts of the article.

Answear:

Honorable Mr. Reviewer,

Initially we sincerely appreciate your help for your constructive review of the article. Your comments are extremely valuable and were an excellent contribution to improving this study. We would like to inform you that we have made improvements that address the concerns you have indicated. We fully understand the highlighted important aspects such as the lack of consistency and readability of the review and the occurrence of spelling errors or omissions, which, after extremely careful review, changes have been applied to improve the article on the issues raised. We are grateful for your detailed observations, which certainly helped to improve our work.

We are convinced that thanks to your comments, our work has become more complete and valuable and has acquired a higher scientific value.

Please accept my deepest thanks for your time and valuable opinion.

Reviewer 3 Report

Comments and Suggestions for Authors

In this manuscript, authors stated, by analyzing the research articles from recent years,  the phytotherapy could be a promising therapeutic solution to treat vulvovaginal candidiasis (VVC) that is mainly caused by Candida albicans species. This alternative method could reduce the strains resistance to the existing treatments including a local or oral preferential therapy (e.g., fluconazole). This narrative review could provide the insights of the comparison of classical pharmacotherapy and herbal therapy in the treatment of VVC, potentially contributing to the care of gynecological patients. Before considering to publish it, I may have some concerns:

1.      Figure1 mentioned in Ln 90-91 and the sentence of “Flowchat below presents paper slection.” could be deleted and authors can add several sentences to briefly introduce how many papers were selected from database and discussed in this manuscript.

2.      Figure 2 in the manuscript should be renumbered to “Figure 1”, so that the description of “The figure number 1 shown below shows chronologically the process of patho-106 genesis of candidiasis.” (Ln 106-107) can be easily understood, otherwise, it may confuse the reader.

3.      The color and size of fonts from Figure 2 and Figure 3 should be changed to highlight that information from the blue background, which is a little bit difficult to tell.

4.      From the main text, authors use “Allium yesdianum”, which should be “Allium jesdianum”? Please check it again. Could author further explain the MIC values for Table number 1? What is the conclusion for Comparison of MIC for fluconazole and Allium jesdianum?

5.      Please check the unit of “ug/ml/ml” (Ln 223, 224), which is correct?

6.      Please check Table number 2, the description of “Number of active C. albicans cells in biofilms [%]” should be corrected to “Number of active C. albicans cells decreased in biofilms [%]”?

7.      Ln 313, is “25 25 ug/ml” correct?

8.      From Table 4 and Table 5, the format of percentage should be checked? For example, “35,71%” should be corrected to “35.71%”?

9.      In Table 6, the information from the column of “Searching group” and “Main findings” could be rewritten by a shorter sentence. And the space between “Authors” and “Years”, “Years” and “Substance” could be narrowed more to ensure the whole table could be in one page for the easy reading.

10. Reference 24, 27, 41, 58, 79, need to be double checked or replaced by some other articles.

11. Some typos should be corrected accordingly. 

Good luck~

Author Response

Manuscript: ijms - 2926187

Response to the reviewers

Dear Reviewer,

First and foremost, thank you for your time that You dedicated to prepare the comments to our manuscript entitled: "Phytotherapy of vulvovaginal candidiasis - a narrative review”. We consider all of the comments very valuable and helpful for the improvement of our paper. We have studied each comment very carefully and made appropriate corrections, which we hope will meet Your approval. Main corrections in the paper and responses to the reviewer’s comments are listed below.

On Behalf of the authors,

Małgorzata Satora

#Review 3 - The corrected passages, as suggested, are highlighted in red in the manuscript.

1.  Figure1 mentioned in Ln 90-91 and the sentence of “Flowchat below presents paper selection.” could be deleted and authors can add several sentences to briefly introduce how many papers were selected from database and discussed in this manuscript.

Answear: Thank you very much for your comment! As recommended, we have removed the sentence “Flowchat below presents paper selection.” and figure 1. The data it contained has been entered into the section titled "Materials and methods" (Ln 82 - 85).

2.  Figure 2 in the manuscript should be renumbered to “Figure 1”, so that the description of “The figure number 1 shown below shows chronologically the process of patho-106 genesis of candidiasis.” (Ln 106-107) can be easily understood, otherwise, it may confuse the reader.

Answear: Thank you for your valuable comment! We have made the recommended corrections that will not confuse the reader. (Ln 109)

3.  The color and size of fonts from Figure 2 and Figure 3 should be changed to highlight that information from the blue background, which is a little bit difficult to tell.

Answear: Thank you! Of course, the color and size of fonts of a Figures 1 and 2 have been changed to make them more readable. (Ln 108 and Ln 129)

4. From the main text, authors use “Allium yesdianum”, which should be “Allium jesdianum”? Please check it again. Could author further explain the MIC values for Table number 1? What is the conclusion for Comparison of MIC for fluconazole and Allium jesdianum?

Answear: Thank you for your observation! After delving into the literature again, we corrected all the names to Allilum jesdianum (Ln 26, 202, 205, 207 - 208, 210). Naturally, the MIC's explanation is included (Ln 205) and the conclusions drawn from Table 1 are added in color green (Ln 209,212).

5.  Please check the unit of “ug/ml/ml” (Ln 223, 224), which is correct?

Answear: Thank you! Definitely the unit is ug/mL, thank you for correcting the oversight (Ln 225 - 226).

6.  Please check Table number 2, the description of “Number of active C. albicans cells in biofilms [%]” should be corrected to “Number of active C. albicans cells decreased in biofilms [%]”?

Answear: Thank you for your valuable comment! Naturally, we took note of the comment and added the word "decreased" (Table 2).

7.   Ln 313, is “25 25 ug/ml” correct?

Answear: Thank you for your vigilance! The value, naturally, was 25 ug/ml. The error has been corrected (Ln 315).

8.   From Table 4 and Table 5, the format of percentage should be checked? For example, “35,71%” should be corrected to “35.71%”?

Answear: Thank you for the right comment! All commas have been changed to periods (Table 4 and 5).

9.  In Table 6, the information from the column of “Searching group” and “Main findings” could be rewritten by a shorter sentence. And the space between “Authors” and “Years”, “Years” and “Substance” could be narrowed more to ensure the whole table could be in one page for the easy reading.

Answear: Thank you for your comment! Of course, we have improved the table, making it more readable and understandable, and according to other recommendations, it fits on one page (Page 11).

10. Reference 24, 27, 41, 58, 79, need to be double checked or replaced by some other articles.

Answear: Thank you! All references listed have been replaced by more qualified medical articles (24: Ln 518 - 520; 27: Ln 524 - 525; 41: Ln 557 - 558; 58: Ln 604 - 606; 79: Ln 658 - 660).

11. Some typos should be corrected accordingly.

Answear: Thank you for your observations! Naturally, all spelling errors have been corrected.

Ln 39, 106, 128, 133, 184 - 185, 192, 217, 267, 273, 335 - 336, 396, 428.

Round 2

Reviewer 2 Report

Comments and Suggestions for Authors

No comments.

Reviewer 3 Report

Comments and Suggestions for Authors

I think the authors has improved the manuscript, which should be good for the publication. 

Thanks~